# A Transcriptomic Benchmark for Foundation Models in Immunology and Inflammation Drug Development

**Karim El Kanbi, Yannis Cattan, Pierre Marschall, Matthew Corney,**
**Aziz Fouché, Vincent Bouget, Julien Duquesne**

Scienta Lab, Paris

## Abstract

Foundation models for transcriptomics are increasingly evaluated on technical metrics disconnected from drug development. We introduce an immunology and inflammation (I&I) benchmark of 35 tasks across 8 diseases, organized along the drug development pipeline: target discovery, preclinical translation, and clinical applications. Tasks span treatment response, clinical severity, molecular perturbations, and patient endotypes, with cross-species, cross-disease, and cross-platform transfer to test translational generalization. Patient sample sizes range from 9 to 713, reflecting data-limited regimes typical of early clinical research. We evaluate general-purpose and domain-specific foundation models against statistical baselines. Foundation models achieve the largest gains on translational tasks (perturbation prediction and cross-species transfer) where baselines fail. Treatment outcome prediction and patient stratification also favor foundation models, while clinical severity prediction remains competitive with feature-selected regression. A domain-specific model (EVA) pretrained on I&I data outperforms general-purpose models across most task categories. Benchmark performance improves with pretraining steps without saturating, suggesting it can serve as a diagnostic for model development.

## 1 Introduction

Biological foundation models have emerged as a promising paradigm for learning representations from large-scale molecular and imaging data (Theodoris, 2024), with notable contributions in transcriptomics (Cui et al., 2024; Theodoris et al., 2023; Hao et al., 2024), histology (Filiot et al., 2023; Chen et al., 2024), genomics (Dalla-Torre et al., 2025; Nguyen et al., 2024), and proteins (Lin et al., 2023; Hayes et al., 2025). Yet evaluation practices remain fragmented: single-cell models are typically assessed on technical metrics like batch integration and cell clustering, while histology models focus predominantly on oncology classification tasks. Recent benchmarks reveal that these models often fail to outperform simple baselines on relevant downstream applications (Kedzierska et al., 2025; Ahlmann-Eltze et al., 2025), exposing a disconnect between pretraining objectives and the representations required for drug development.

Standardized benchmarks have catalyzed progress in other domains (ImageNet for computer vision (Russakovsky et al., 2015), GLUE for natural language processing (Wang et al., 2018)), yet the biological foundation model field lacks consensus evaluation protocols for translational research. This gap is particularly acute in immunology and inflammation (I&I), a therapeutic area where shared pathogenic mechanisms across diseases (Lincoln et al., 2024; González-Serna et al., 2020) create unique opportunities for transfer learning, but where evaluation has been ad hoc and inconsistent.

We introduce an I&I benchmark of 35 tasks spanning 8 diseases and the full drug development pipeline. The benchmark follows four design principles: (i) **drug development relevance** (with tasks curated by immunologists with drug development expertise, selecting endpoints based on regulatory relevance and availability of paired molecular-clinical data); (ii) **diverse prediction paradigms** (supervised and zero-shot, across treatment response, clinical severity, perturbations, en-

dotypes); (iii) **translational generalization** (cross-species, cross-disease, cross-platform transfer); and (iv) **methodological rigor** (subject-level splitting, five-seed evaluation, appropriate baselines). Patient sample sizes range from 9 to 713, reflecting real clinical conditions. We evaluate current foundation models and reveal where they succeed, where they fail, and what this implies for future development.

## 2 BENCHMARK DESIGN

### 2.1 TASK TAXONOMY

The benchmark organizes 35 tasks into three categories aligned with the drug development pipeline (Table 1). Tasks span 8 diseases: atopic dermatitis (AD), psoriasis (Pso), hidradenitis suppurativa (HS), Sjögren's disease (SjD), rheumatoid arthritis (RA), and inflammatory bowel disease (IBD), comprising Crohn's disease (CD) and ulcerative colitis (UC).

**Discovery.** Two task types evaluate early-stage drug development. *Zero-shot target efficacy prediction* tests whether perturbing a drug's molecular target shifts patient transcriptomes toward healthy states, using decoder gradients for *in silico* gene perturbations (Bjerregaard et al., 2025), spanning 114 drug-disease combinations across 8 diseases. *Gene function prediction* assesses whether gene embeddings capture functional relationships through multi-label classification of disease associations, GO terms, cell type markers, and pathway membership.

**Preclinical.** Two task types address the translation bottleneck between model organisms and human disease. *Cross-species treatment effect prediction* trains on mouse perturbation data and evaluates on human samples using ortholog mapping for fair comparison. *Molecular perturbation prediction* assesses full transcriptomic response prediction, including cross-disease transfer tasks testing whether perturbation signatures generalize across indications sharing the same drug target.

**Clinical.** Three task types span patient-facing applications. *Molecular to clinical activity* predicts clinical severity indices from gene expression across 6 diseases (14 tasks). *Stratification into endotypes* classifies rheumatoid arthritis pathotypes from blood or synovial tissue. *Treatment outcome prediction* evaluates binary therapeutic response prediction in IBD, including cross-treatment and cross-platform generalization tasks.

### 2.2 EVALUATION PROTOCOL

All tasks employ subject-level data splitting to prevent leakage from repeated measurements. We report results across five random seeds. Expression data undergo $\log_2(\text{CPM} + 1)$ normalization.

**Statistical baselines.** We compare foundation models against task-appropriate statistical baselines: (i) *regression tasks*: ridge regression with K-best feature selection (k=8000); (ii) *classification tasks*: logistic regression with K-best feature selection (k=8000); (iii) *perturbation prediction*: average perturbation predictor (predicting the training set mean expression change); (iv) *zero-shot target efficacy*: gene interaction matrix estimated via linear regression from observational data; (v) *gene function prediction*: PCA embeddings from an I&I bulk RNA-seq dataset. These baselines provide principled references for assessing foundation model utility.

**Adaptation strategies.** Beyond zero-shot tasks, we support linear probing (frozen encoder), LoRA, and last-layer fine-tuning. The appropriate strategy can be selected per task type depending on sample size and task complexity.

**Metrics.** Classification tasks report AUROC; multi-label classification (gene function) reports AUPRC; regression tasks report Pearson correlation; perturbation tasks report Pearson correlation on expression changes (post-treatment minus baseline).

Table 1: Benchmark tasks (35 total). N denotes number of samples (patients or paired observations for perturbation tasks).

| Stage | Task Type | Task | Disease | Tissue | N |
|---|---|---|---|---|---|
| **Discovery** | Zero-shot (1) | Target efficacy | 8 diseases | Various | 114 pairs |
| | Gene function (5) | Disease associations | I&I (6) | – | 19K |
| | | GO terms | – | – | 19K |
| | | Cell type markers | – | – | 19K |
| | | Reactome pathways | – | – | 19K |
| | | WikiPathways | – | – | 19K |
| **Preclinical** | Cross-species treatment effect (2) | Dupilumab | AD | Skin | 338 |
| | | TNFi | RA | Synovial | 36 |
| | Molecular perturbation (5) | Adalimumab HS→Pso | HS/Pso | Skin | 14/17 |
| | | Adalimumab Pso→HS | Pso/HS | Skin | 17/14 |
| | | Anti-TNF | IBD | Mouse colon | 9 |
| | | Rituximab | SjD | Blood | 68 |
| | | Rituximab | SjD | Salivary | 11 |
| **Clinical** | Molecular to clinical activity (14) | TJC28 | RA | Synovial | 345 |
| | | SJC28 | RA | Synovial | 165 |
| | | SES-CD | CD | Digestive | 696 |
| | | GHAS-7 | CD | Digestive | 460 |
| | | HBI | CD | Digestive | 696 |
| | | PASI | Pso | Skin | 92 |
| | | SCORAD | AD | Skin | 207 |
| | | EASI | AD | Skin | 238 |
| | | Nancy Index | UC | Digestive | 104 |
| | | SCCAI | UC | Digestive | 249 |
| | | Endoscopic Mayo | UC | Digestive | 249 |
| | | IgA | SjD | Blood | 576 |
| | | IgG | SjD | Blood | 713 |
| | | ESSDAI Bio | SjD | Blood | 612 |
| | Stratification into endotypes (2) | RA pathotype | RA | Blood | 59 |
| | | RA pathotype | RA | Synovial | 270 |
| | Treatment outcome (6) (endoscopic remission) | Adalimumab | IBD | Digestive | 63 |
| | | Infliximab | IBD | Digestive | 35 |
| | | IFX→ADA | IBD | Digestive | 98 |
| | | ADA→IFX | IBD | Digestive | 98 |
| | | Vedolizumab RNAseq→microarray | IBD | Digestive | 165 |
| | | Vedolizumab microarray→RNAseq | IBD | Digestive | 165 |

## 3 RESULTS

We evaluated transcriptomic foundation models scGPT (Cui et al., 2024) and BulkRNABert (Gélard et al., 2025), alongside EVA, a domain-specific foundation model we developed, pretrained on I&I transcriptomics data across human and mouse. EVA's pretraining corpus does not include any of the benchmark downstream task datasets or labels. We compare these models against statistical baselines (Table 2). Values are macro-averages over tasks within each category, computed from 5-seed means per task. For treatment outcome, we retained only tasks where at least one model exceeded 0.6 AUROC, ensuring sufficient signal for meaningful comparison.

**Foundation models excel at translational tasks.** Cross-disease and cross-species tasks show clear advantages for foundation models over statistical baselines. Molecular perturbation prediction achieves 0.547 vs. 0.171 Pearson on average, with the largest gains on cross-disease transfer where baselines fail entirely (e.g., adalimumab HS→Pso: 0.45 vs. −0.19). Cross-species treatment effect prediction similarly favors foundation models (0.445 vs. 0.025). Zero-shot target efficacy (0.693 vs. 0.569 AUROC) confirms that foundation models capture transferable biological signatures that simple feature selection cannot recover.

**Clinical prediction tasks remain competitive with baselines.**    Molecular to clinical activity prediction shows no clear foundation model advantage on average (0.434 vs. 0.427 Pearson), with heterogeneous results across endpoints (Table S1). Treatment outcome and stratification tasks favor foundation models on aggregate, but gains are modest on some individual tasks. These results highlight where current pretraining may not yet align with clinical prediction targets, consistent with recent findings (Ahlmann-Eltze et al., 2025).

Table 2: Benchmark results (macro-averaged over tasks, 5 seeds per task). Bold: best; underline: second-best. BulkRNABert cannot perform zero-shot prediction as the decoder is unavailable.

| Stage | Task | Metric | EVA | scGPT | BulkRNABert | Stat. Baseline |
|---|---|---|---|---|---|---|
| Discovery | Zero-shot target efficacy | AUROC | **0.693** | 0.539 | – | 0.569 |
| | Gene function | AUPRC | **0.494** | 0.357 | 0.287 | 0.328 |
| Preclinical | Molecular perturbation | Pearson | **0.547** | 0.454 | 0.475 | 0.171 |
| | Cross-species treatment effect | Pearson | **0.445** | 0.439 | 0.435 | 0.025 |
| Clinical | Molecular to clinical activity | Pearson | **0.434** | 0.360 | 0.312 | 0.427 |
| | Stratification into endotypes | AUROC | **0.786** | 0.706 | 0.695 | 0.762 |
| | Treatment outcome | AUROC | **0.650** | 0.491 | 0.497 | 0.604 |

**The benchmark tracks pretraining improvements.**    Performance improves with pretraining steps across task categories (Figure 1), demonstrating that the benchmark is sensitive to model development choices. Notably, scaling curves show no sign of saturation, suggesting foundation models have substantial room for improvement on these tasks. This positions the benchmark as a diagnostic for foundation model training, revealing which capabilities emerge with continued pretraining and where current approaches plateau.

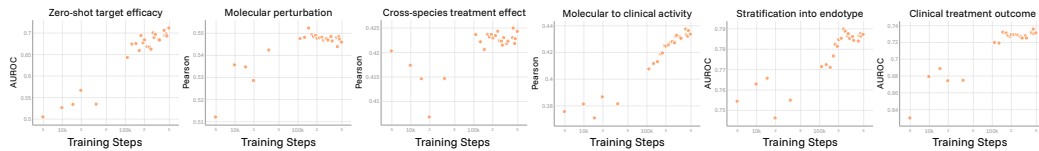

Figure 1: Benchmark performance improves with training steps across evaluation categories. Each panel shows EVA performance on different task groups as a function of pretraining steps, demonstrating that pretraining improvements consistently transfer to downstream tasks.

## 4    DISCUSSION

Our benchmark reveals a nuanced picture of biological foundation model capabilities. The clearest successes (treatment outcome, stratification) involve tasks requiring complex phenotypic signatures, while perturbation prediction shows strong gains primarily on cross-disease transfer tasks, suggesting models capture transferable but not necessarily causal signatures. The benchmark enables systematic evaluation across axes that matter for drug development: cross-species, cross-disease, and cross-platform transfer. Sample size diversity (9 to 713) tests models in data-limited regimes typical of early-phase trials. The consistent advantage of domain-specific pretraining (EVA) over generic models suggests therapeutically relevant data may be more valuable than scale alone.

Several limitations merit discussion. The benchmark currently focuses on I&I, and generalization to other therapeutic areas remains to be established. The benchmark supports multiple adaptation strategies (linear probing, LoRA, last-layer fine-tuning); choice of strategy may affect relative model rankings. This benchmark is a work in progress and will be made publicly available to enable community-driven evaluation of biological foundation models on clinically relevant tasks.

MEANINGFULNESS STATEMENT

This benchmark helps us learn meaningful representations of life by revealing which model architectures and training strategies capture biology that matters for human health. By evaluating models on cross-species transfer, cross-disease generalization, and diverse clinical outcomes rather

than technical metrics, we identify representations that encode conserved disease mechanisms and therapeutic responses. The benchmark's translational tasks expose when models learn superficial correlations versus causal relationships, while sample size diversity tests robustness under real clinical constraints. Results showing domain-specific models outperform generic foundation models suggest that meaningful representations require pretraining on therapeutically relevant data, guiding model development toward clinically actionable biological understanding.

## LLM USAGE POLICY

We used Claude (Anthropic) as a coding assistant to help write and debug experimental code and data processing scripts. For manuscript preparation, we used Claude and Grammarly to improve grammar, clarity, and writing quality. All research ideas, experimental design, scientific claims, and results were developed independently by the authors. All LLM-generated code and text were reviewed, validated, and verified by the authors.

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

## SUPPLEMENTARY MATERIAL

Table S1: Detailed benchmark results by task (mean $\pm$ std over 5 seeds). Metrics: AUROC (zero-shot target efficacy, stratification, treatment outcome), AUPRC (gene function), Pearson correlation (perturbation, cross-species, mol. to clinical). Bold: best; underline: second-best. BulkRNABert cannot perform zero-shot prediction as the decoder is unavailable.

| Task | EVA | scGPT | BulkRNABert | Baseline |
|---|---|---|---|---|
| **Zero-shot target efficacy (AUROC)** | | | | |
| Target efficacy | **0.69** | 0.54 | – | 0.57 |
| **Gene function (AUPRC)** | | | | |
| CellType | $0.41 \pm 0.19$ | $\mathbf{0.48 \pm 0.24}$ | $0.34 \pm 0.21$ | $\underline{0.39 \pm 0.23}$ |
| Disease | $\mathbf{0.46 \pm 0.34}$ | $\underline{0.24 \pm 0.32}$ | $0.22 \pm 0.26$ | $0.25 \pm 0.30$ |
| GO | $\mathbf{0.47 \pm 0.27}$ | $\underline{0.34 \pm 0.28}$ | $0.30 \pm 0.22$ | $0.29 \pm 0.28$ |
| Reactome | $\mathbf{0.64 \pm 0.29}$ | $\underline{0.36 \pm 0.22}$ | $0.30 \pm 0.23$ | $0.32 \pm 0.25$ |
| WikiPathways | $\mathbf{0.49 \pm 0.27}$ | $\underline{0.37 \pm 0.21}$ | $0.27 \pm 0.20$ | $0.38 \pm 0.23$ |
| **Molecular perturbation (Pearson)** | | | | |
| Anti-TNF (IBD mice) | $\mathbf{0.70 \pm 0.16}$ | $0.37 \pm 0.19$ | $0.49 \pm 0.18$ | $\underline{0.59 \pm 0.21}$ |
| Adalimumab (HS $\rightarrow$ Pso) | $\mathbf{0.45 \pm 0.01}$ | $\underline{0.41 \pm 0.02}$ | $0.41 \pm 0.03$ | $-0.19 \pm 0.02$ |
| Adalimumab (Pso $\rightarrow$ HS) | $\mathbf{0.32 \pm 0.00}$ | $\underline{0.23 \pm 0.01}$ | $0.24 \pm 0.02$ | $-0.13 \pm 0.00$ |
| Rituximab (SjD blood) | $\mathbf{0.77 \pm 0.02}$ | $\underline{0.75 \pm 0.04}$ | $0.74 \pm 0.02$ | $0.60 \pm 0.04$ |
| Rituximab (SjD salivary) | $0.50 \pm 0.10$ | $\mathbf{0.51 \pm 0.10}$ | $\underline{0.49 \pm 0.12}$ | $-0.01 \pm 0.09$ |
| **Cross-species treatment effect (Pearson)** | | | | |
| Dupilumab | $\underline{0.48 \pm 0.00}$ | $0.47 \pm 0.00$ | $\mathbf{0.48 \pm 0.00}$ | $0.05 \pm 0.00$ |
| TNFi RA | $\underline{0.41 \pm 0.00}$ | $\mathbf{0.41 \pm 0.00}$ | $0.39 \pm 0.00$ | $0.00 \pm 0.00$ |
| **Molecular to clinical activity (Pearson)** | | | | |
| Blood IgA | $0.37 \pm 0.11$ | $0.28 \pm 0.13$ | $0.18 \pm 0.12$ | $\mathbf{0.39 \pm 0.10}$ |
| Blood IgG | $\underline{0.51 \pm 0.10}$ | $0.41 \pm 0.06$ | $0.25 \pm 0.15$ | $\mathbf{0.54 \pm 0.09}$ |
| Digestive GHAS-7 | $0.57 \pm 0.02$ | $\underline{0.58 \pm 0.05}$ | $0.55 \pm 0.07$ | $\mathbf{0.61 \pm 0.03}$ |
| Digestive SES-CD | $\mathbf{0.46 \pm 0.09}$ | $0.39 \pm 0.07$ | $0.35 \pm 0.07$ | $\underline{0.44 \pm 0.08}$ |
| ESSDAI Bio | $\mathbf{0.41 \pm 0.16}$ | $0.33 \pm 0.08$ | $0.13 \pm 0.15$ | $\underline{0.40 \pm 0.08}$ |
| Endoscopic Mayo | $0.64 \pm 0.15$ | $\underline{0.65 \pm 0.15}$ | $0.60 \pm 0.15$ | $\mathbf{0.66 \pm 0.12}$ |
| HBI | $\mathbf{0.24 \pm 0.07}$ | $0.14 \pm 0.06$ | $0.12 \pm 0.08$ | $\underline{0.19 \pm 0.06}$ |
| Nancy Histological Index | $0.73 \pm 0.09$ | $\underline{0.75 \pm 0.09}$ | $0.71 \pm 0.14$ | $\mathbf{0.76 \pm 0.12}$ |
| RA SJC28 | $\mathbf{0.44 \pm 0.16}$ | $0.29 \pm 0.20$ | $0.24 \pm 0.18$ | $\underline{0.36 \pm 0.20}$ |
| RA TJC28 | $0.27 \pm 0.17$ | $0.24 \pm 0.20$ | $0.19 \pm 0.21$ | $\mathbf{0.28 \pm 0.25}$ |
| SCCAI | $\mathbf{0.59 \pm 0.07}$ | $0.50 \pm 0.06$ | $0.49 \pm 0.09$ | $\underline{0.57 \pm 0.05}$ |
| Skin EASI | $\underline{0.29 \pm 0.20}$ | $0.29 \pm 0.16$ | $0.24 \pm 0.15$ | $\mathbf{0.30 \pm 0.22}$ |
| Skin PASI | $\mathbf{0.26 \pm 0.24}$ | $0.10 \pm 0.16$ | $0.21 \pm 0.08$ | $\underline{0.25 \pm 0.22}$ |
| Skin SCORAD | $\mathbf{0.27 \pm 0.14}$ | $0.07 \pm 0.13$ | $0.09 \pm 0.11$ | $\underline{0.21 \pm 0.19}$ |
| **Stratification into endotypes (AUROC)** | | | | |
| RA blood | $\mathbf{0.65 \pm 0.12}$ | $0.52 \pm 0.07$ | $0.58 \pm 0.08$ | $\underline{0.63 \pm 0.12}$ |
| RA joint | $\mathbf{0.92 \pm 0.01}$ | $\underline{0.89 \pm 0.02}$ | $0.81 \pm 0.03$ | $0.90 \pm 0.01$ |
| **Treatment outcome (Endoscopic remission) (AUROC)** | | | | |
| IBD, adalimumab | $\mathbf{0.76 \pm 0.07}$ | $0.40 \pm 0.12$ | $0.37 \pm 0.16$ | $\underline{0.48 \pm 0.17}$ |
| IBD, infliximab | $\mathbf{0.80 \pm 0.24}$ | $0.50 \pm 0.13$ | $0.48 \pm 0.18$ | $\underline{0.62 \pm 0.10}$ |
| IBD, ADA $\rightarrow$ IFX | $0.50 \pm 0.05$ | $0.59 \pm 0.05$ | $\underline{0.64 \pm 0.07}$ | $\mathbf{0.66 \pm 0.04}$ |
| IBD, IFX $\rightarrow$ ADA | $0.57 \pm 0.04$ | $0.61 \pm 0.04$ | $\underline{0.57 \pm 0.02}$ | $\mathbf{0.62 \pm 0.02}$ |
| IBD, vedolizumab, seq $\rightarrow$ arr | $\mathbf{0.65 \pm 0.04}$ | $0.47 \pm 0.09$ | $0.47 \pm 0.07$ | $\underline{0.61 \pm 0.01}$ |
| IBD, vedolizumab, arr $\rightarrow$ seq | $0.63 \pm 0.05$ | $0.37 \pm 0.03$ | $0.44 \pm 0.07$ | $\mathbf{0.64 \pm 0.02}$ |

