# OpenReview forum: "A Transcriptomic Benchmark for Foundation Models in Immunology and Inflammation Drug Development"
_ICLR.cc/2026/Workshop/LMRL — ICLR 2026 Workshop LMRL Poster_

### Official Review · Reviewer_6Smg · 2026-02-13
**Strong, rigorous benchmark for immunology and inflammation foundation models, perhaps some minor clarification needed**

**Rating:** 8
**Confidence:** 4

**Review:**

The authors highlight a lack of standardised benchmarking in biological foundation model development and aim to improve this in the specific field of immunology and inflammation. They develop and present a thorough benchmark spanning 35 tasks across 3 diseases that aims to cover the whole drug discovery pipeline from discovery, preclinical and clinical. They apply these benchmarks to general purpsoe models a domain specific model and selected statistical baselines. They demonstrate that the benchmark effectively evaluates these models and predicts the efficacy of pretraining.

I think this work addresses an important problem and has been done in a thorough and rigorous way to create a meaningful benchmark, the kind of which is needed in biology. On the whole the work is presented clearly and as someone who is not in drug development or immunology and inflammation I feel I could understand the results presented, even if I don't have prior knowledge of all the specific context. As someone not in the fields I have a couple of minor points of confusion. Firstly, the EVA model is presented but it is not clear if this was trained by the authors or if it is a previously existing model. Secondly, I do not have a concrete idea of the structure of the tasks in the benchmark i.e. what are the inputs and outputs for each task, if and how models are trained during the tasks etc. Perhaps this can be expanded upon in an appendix.

Overall I would strongly recommend acceptance of this paper.

Pros:
- Addresses an important and timely problem: lack of standardized benchmarking in biological foundation models.
- Benchmark is meaningful, spanning 35 tasks, 3 diseases, and covering the full drug discovery pipeline
- Demonstrates the benchmark is effective at discriminating model performance and predicting pretraining efficacy
- Work is presented clearly and is understandable even to non-experts outside drug development/immunology

Cons
- Unclear whether the EVA model is newly trained by the authors or a previously existing model
- Benchmark task details are sometimes not fully explained (inputs/outputs, training setup per task, whether models are fine-tuned, etc.)

---

### Official Review · Reviewer_saDF · 2026-02-24
**The authors introduced a new transcriptomic benchmark focused on immunology and inflammation, covering several tasks and diseases relevant to the drug development pipeline. Additionally, they evaluated foundation models and statistical baselines and reported some preliminary conclusions, such as foundation models showing the largest advantages in translational tasks, while traditional baselines remain competitive in clinical predictions.**

**Rating:** 7
**Confidence:** 3

**Review:**

Summary:
The authors introduced a new transcriptomic benchmark focused on immunology and inflammation, covering several tasks and diseases relevant to the drug development pipeline. Additionally, they evaluated foundation models and statistical baselines and reported some preliminary conclusions, such as foundation models showing the largest advantages in translational tasks, while traditional baselines remain competitive in clinical predictions.
The work addresses an important problem of evaluating foundation models on drug development relevant tasks. I have some comments and suggestions in order to improve this work:

1) It was not so clear. Perhaps the authors should state whether EVA was pretrained on datasets overlapping with the benchmark tasks.
2) Regarding baseline comparisons, while statistical baselines (e.g., ridge regression, PCA embeddings) are included, they are relatively simple. More sophisticated baselines, such as ensemble methods or domain adapted traditional machine learning models, could provide a stronger point of comparison. For example, how do foundation models compare to state of the art feature engineering pipelines or hybrid models?
3) The authors mention, "All tasks employ subject level data splitting to prevent leakage from repeated measurements. We report results across five random seeds. Expression data undergo log2(CPM + 1) normalization." I am double checking: was the expression data normalization performed within folds, or was it done first before devising the folds? I want to confirm the potential for data leakage.

---

### Official Review · Reviewer_JLbY · 2026-02-24
**A novel, clinically motivated benchmark, strong design but could use more models**

**Rating:** 8
**Confidence:** 3

**Review:**

## Summary —
This work introduces a novel benchmark that is framed around the drug development pipeline, with a focus on immunology and inflammation (I&I). Commonly used transcriptomic foundational model metrics are disconnected from their useful, clinically relevant applications.

## Strengths —
Subject-level splitting to prevent data leakage, five-seed evaluation, and appropriate task-specific baselines are all strong choices.

The sample size diversity (9 to 713 patients) is practical; in clinical settings where data may be severely limited.

Another pre-print, (Elmarakeby, 2025) somewhat mirrors this one but with an oncology focus. Suggesting there is growing interest in benchmarking domain specific applications of scFMs. These results will serve I&I focused researchers well--and might even motivate future pathology specific benchmarks.

## Weaknesses —
While the task selection is broad, and probably does capture the bottlenecks in a drug development pipeline. A justification/citation of the choices would strengthen this paper beyond “a domain expert”: L50  “(i) drug development relevance (with tasks curated alongside a domain expert to ensure biological plausibility…”

Model selection - some justification for the model choices would strengthen this paper. Notably, why are commonly cited models like CellFM and Geneformer not included? Ideally model selection would be informed by prior benchmarks. Elmarakeby, 2025 uses many more foundation models (Geneformer, scFoundation, CellPLM, etc…)

Additionally, it is a bit odd to build a benchmark around I&I tasks, yet only use one model that is I&I focused. Though, from my search, EVA is the only model with this I&I focus. Ideally fine-tuning the other models would make this comparison more robust.

## Questions for authors —
Baselines - Is there any reason for not including a simple MLP? For ridge and logistic regression, why is k=8000, i.e why not more/less?

Do the authors think an overall diagram would help? With the breadth of tasks, I think it might be easier to interpret than the current Table 1.

Again, some justification for the chosen tasks would help, especially for someone not familiar with I&I.

## Recommendation —
Accept. Strong paper which introduces novel metrics for benchmarking models in clinically relevant applications. Though, the choice of models could probably be expanded given the focus on I&I.

---

### Meta-Review · Area_Chair_w7Uw · 2026-02-25

**Recommendation:** Accept (Poster)
**Confidence:** 4

**Metareview:**

Accept

---

### Decision · Program_Chairs · 2026-03-02

**Decision:**

Accept (Spotlight)

**Comment:**

Please see the meta-review.